# Validation Study of Novel Point-of-Care Tests for Infliximab, Adalimumab and C-Reactive Protein in Capillary Blood and Calprotectin in Faeces in an Ambulatory Inflammatory Bowel Disease Care Setting

**DOI:** 10.3390/diagnostics13101712

**Published:** 2023-05-12

**Authors:** Adriaan Volkers, Mark Löwenberg, Marlou Braad, Yara Abeling, Krisztina Gecse, Nicole Berkers, Nahid Montazeri, Geert D’Haens

**Affiliations:** 1Department of Gastroenterology and Hepatology, Amsterdam University Medical Center, Meibergdreef 9, 1105 AZ Amsterdam, The Netherlands; 2Biostatistics Unit, Department of Gastroenterology and Hepatology, Amsterdam University Medical Center, Meibergdreef 9, 1105 AZ Amsterdam, The Netherlands

**Keywords:** point-of care, inflammatory bowel diseases, follow-up, anti-tumour necrosis factor, biomarkers, Crohn’s disease, ulcerative colitis

## Abstract

**Background and aims:** Point-of-care tests (POCT) allow instant measurement of inflammatory markers and drug concentrations. Here, we studied agreement between a novel POCT device and reference methods of measuring infliximab (IFX) and adalimumab (ADL) serum concentrations and C-reactive protein (CRP) and faecal calprotectin (FCP) concentrations of patients with inflammatory bowel disease (IBD). **Methods:** In this single-centre validation study, IBD patients were recruited in which IFX, ADL, CRP and/or FCP tests were required. IFX, ADL and CRP POCT were performed on capillary whole blood (CWB), obtained via finger prick. Additionally, IFX POCT was performed on serum samples. FCP POCT was performed on stool samples. Agreement between POCT and reference methods was tested using Passing–Bablok regression, intra-class correlation coefficients (ICC) and Bland–Altman plots. **Results:** In total, 285 patients participated. Passing–Bablok regression identified differences between the reference method and IFX CWB POCT (intercept = 1.56), IFX serum POCT (intercept = 0.71, slope = 1.10) and ADL CWB POCT (intercept = 1.44). There were also differences in the Passing–Bablok regressions of CRP (intercept = 0.81, slope = 0.78) and FCP (intercept = 51 and slope = 0.46). Bland–Altman plots demonstrated that IFX and ADL concentrations were slightly higher with the POCT and CRP and FCP were slightly lower with POCT. The ICC demonstrated almost perfect agreement with IFX CWB POCT (ICC = 0.85), IFX serum POCT (ICC = 0.96), ADL CWB POCT (ICC = 0.82) and CRP CWB POCT (ICC = 0.91) and moderate agreement with FCP POCT (ICC = 0.55). **Conclusions:** IFX and ADL results were slightly higher with this novel rapid and user-friendly POCT, whereas CRP and FCP results were slightly lower compared to the reference methods.

## 1. Introduction

Ileocolonoscopy is the gold standard for assessing disease activity in inflammatory bowel disease (IBD) patients [1]. However, due to cost and patient burden ileocolonoscopy cannot be used frequently. Faecal calprotectin (FCP) and C-reactive protein (CRP) are among the best-studied non-invasive markers of inflammation in IBD [2]. In particular, FCP has been shown to correlate well with endoscopic disease activity [3]. In the CALM study, treatment escalation based on symptoms combined with CRP and FCP in Crohn’s disease (CD) patients led to improved clinical and endoscopic outcomes compared to symptom-driven treatment escalation [4]. Close monitoring with regular CRP and FCP measurements has become standard care in the clinical management of CD and ulcerative colitis (UC) patients.

In addition to inflammatory markers, anti-tumour necrosis factor (TNF) antibody serum concentrations are often measured in IBD patients who are treated with an anti-TNF agent, such as infliximab (IFX) or adalimumab (ADL). Depending on the desired treatment goal, IFX and ADL serum concentration >3 µg/mL and >5 µg/mL, respectively are often being used as therapeutic targets. Based on a recent meta-analysis, reactive therapeutic drug monitoring (TDM) turned out to be beneficial in IBD patients, whereas proactive remains more controversial [5]. However, proactive TDM is often applied in clinical IBD care. Specific indications, such as severe inflammation causing high drug clearance, small bowel or fistulising CD, or acute severe UC, seem to require higher anti-TNF concentrations, justifying a proactive TDM approach [6,7,8].

IFX and ADL serum concentrations results may take up to two weeks, and FCP results may take up to one week. Waiting for these lab results hampers rapid decision-making in daily practice. So-called point-of-care tests (POCT) are diagnostic methods that can be performed at the outpatient setting, enabling rapid treatment decisions. Yet, existing POCT of IFX and ADL require multiple steps and take at least 15 min, making them difficult to implement in clinical practice [9,10]. A novel, more rapid and user-friendly POCT device was recently developed (ProciseDx, San Diego, CA, USA) for IFX, ADL and CRP serum concentrations using capillary whole blood (CWB) and FCP measurements [11,12,13]. The present study aimed to validate this novel POCT device by comparing results with reference methods for IFX, ADL, CRP and FCP in a clinical IBD setting.

## 2. Materials and Methods

### 2.1. Study Design

This was a single-centre investigator-initiated prospective clinical validation study to test for statistical agreement between POCT results and reference lab methods for IFX, ADL, CRP and FCP. The study took place at the IBD outpatient and infusion clinic of the Amsterdam UMC, Location AMC, Amsterdam, The Netherlands. Participants who underwent a finger prick signed informed consent prior to enrolment. The medical ethical committee reviewed the study protocol and concluded that this study did not fall under the scope of the medical scientific research legislation.

Adult (≥18 years) patients with confirmed CD or UC who underwent IFX, ADL, CRP and/or FCP measurements in standard clinical care were eligible. Patients had to be on active treatment with IFX to participate in the IFX validation study or on active treatment with ADL to participate in the ADL validation study. In addition to the conventional IFX, ADL, CRP or FCP measurement (referred to as: ‘reference’ method), the corresponding POCT was performed with a novel POCT device (ProciseDx, San Diego, CA, USA). IFX, ADL and CRP POCT were measured on CWB, IFX was also measured in serum and FCP was measured in faeces. IFX concentrations were either measured as a trough concentration or at an intermediate time point to increase the range of serum IFX results. For the validation of CRP, there had to be a clinical suspicion of active disease according to the physician’s discretion. Results and baseline characteristics were collected on an electronic case report form (Castor EDC, Amsterdam, The Netherlands).

### 2.2. Reference Lab Methods

Serum IFX and ADL serum concentrations were measured with an ELISA developed by Sanquin (Sanquin, Amsterdam, The Netherlands) [14,15]. The lower and upper limits of the IFX quantification ranged from 0.002 µg/mL to 120 µg/mL. The lower limit of the ADL quantification was 0.01 µg/mL and there was no upper limit. A photometric lab assay (Cobas c702 module, Roche, Basel, Switzerland) was used to measure plasma CRP concentrations (referred to as ‘CRP lab assay’). The lower limit of the CRP assay was 0.3 mg/L with no upper limit. FCP concentrations were assessed with an automated enzyme fluoroimmunoassay: Elia Phadia (Phadia 250, Thermo Fisher Scientific, Uppsala, Sweden) with a range between 3.8–6000 mg/kg.

### 2.3. POCT

For CWB IFX, ADL and CRP POCT, a CWB sample of 20 µL was obtained via finger prick with a lancet (2.0 mm BD Microtainer, BD, Franklin Lakes, NJ, USA) and collected in a fixed volume pipette (ProciseDx). The time between venipuncture and finger prick was preferably less than five minutes, with a maximum of 30 min. The CWB sample was dispensed into an IFX, ADL or CRP reaction cartridge (ProciseDx) which contained lyophilised reagent beads, specific for the type of measurement. One buffer bulb with 1.5 mL tris-buffered saline (TBS) was added which was supplied with the cartridges. The cartridge was closed, inverted five times to allow mixing and the reagent beads to dissolve and then placed in the POCT device for subsequent analysis.

In addition to the IFX CWB POCT, an IFX POCT was performed on the serum sample which was used for the IFX ELISA. As the serum samples were stored frozen, the samples were thawed at room temperature and homogenised on a vortex for approximately ten seconds. First, 1 mL TBS was dispensed into an IFX cartridge. In addition, 20 µL of the serum sample was dispensed in the cartridge with a pipette, the cartridge was closed, inverted five times and placed in the POCT device to run the test.

The POCT device uses time-resolved fluorescence resonance energy transfer (TR-FRET) technology to measure the concentrations. The lower and upper limits of the assays were 1.7 μg/mL–77.2 μg/mL for IFX POCT, 1.3 μg/mL–51.5 μg/mL for ADL POCT and 3.6 mg/L–100 mg/L for CRP POCT.

For the FCP sample collection, a stool collection kit (ProciseDx) was used that is developed for home-based patient-use. From the same faecal sample on which the FCP Elia was performed, a sample was obtained by inserting a probe with grooves three to five times into the faeces until the edges of the probe were covered. After two hours of incubation in the sample collector device containing TBS, 200 µL of the dilution was dispensed in the cartridge along with a 1.5 mL TBS. The cartridge was inverted five times and placed into the POCT device. The lower and upper limits of the FCP POCT were 34 mg/kg and 1500 mg/kg, respectively. The entire processing time of the IFX, ADL and FCP POCT was approximately five minutes and CRP POCT took three minutes.

### 2.4. Sample Size

The range ratio was calculated and a guideline for measurement procedure comparison experiments was used to determine the appropriate sample sizes to test for statistical agreement between the reference methods and POCT [16,17]. For the IFX and ADL POCT validation, we aimed to include 120 participants. No sample size was determined for IFX serum POCT, as we aimed to perform this on all samples which were used for the IFX ELISA and for which sufficient residual serum was available. The sample sizes of CRP and FCP POCT were determined at 40 participants having both results within the assay ranges, since these assays were already partly validated [12,13].

### 2.5. Statistical Analysis

Descriptive statistics are presented as means with standard deviations (SD) or medians with interquartile ranges (IQR), if appropriate. Statistical agreement was tested using Passing–Bablok regression with the calculation of the 95% confidence interval (CI) for the slope and intercept of this regression. Here, the agreement was proven if the value ‘0’ was within the 95% CI of the intercept (excluding systemic bias) and if ‘1’ was within the 95% CI of the slope (excluding conditional bias). Agreement was also determined by calculating the intra-class correlation coefficient (ICC, two-way mixed, absolute agreement, single measures) and interpreted as previously described: 0.00–0.20 as ‘slight’, 0.21–0.40 as ‘fair’, 0.41–0.60 as ‘moderate’, 0.61–0.80 as ‘substantial’, 0.81–1.00 as almost perfect [16]. Correlations between POCT results and conventional lab results were calculated with Pearson’s correlation and Spearman’s rank correlation. The agreement was visualised on Bland–Altman plots showing the differences between each test result on the *y*-axis and the mean of both test results on the *x*-axis. The mean difference from all tests (bias) and the upper and lower limit of agreement (95% CI of the bias) are visualised on the Bland–Altman plots. For Passing–Bablok regression, ICC, Bland–Altman plot and Pearson’s correlation, results from patients were included only if both the reference method and POCT result were within their assay ranges. All results could be included in Spearman’s rank correlation. Finally, we calculated discrepancies between the reference methods and POCT with negative predictive values (NPV) and positive predictive values (PPV) by using clinically relevant cut-off values, which were determined at 3 µg/mL for IFX, 5 µg/mL for ADL, 5 mg/L for CRP and 250 mg/kg for FCP. Passing–Bablok regression was executed with R (version 4.2.1, R Development Core Team, Vienna, Austria), Bland–Altman plots were created with Graphpad Prism (version 9.3.1, San Diego, CA, USA) and the other analyses were performed using SPSS (version 28, IBM, Armonk, NY, USA).

## 3. Results

Between June 2020 and December 2021, 285 patients were recruited. The majority were female and had CD (Table 1). For the ADL POCT validation, 92 participants were recruited, of whom 22 were sampled twice and one patient was sampled three times. Out of all serum samples that were used for the IFX ELISA, 79 were available for an additional IFX serum POCT. There was suspicion for timing errors in which the IFX infusion was already running in while the finger prick was being performed. This would have resulted in higher IFX CWB POCT. Therefore, separate analyses are reported in which five IFX CWB POCT results were included.

Passing–Bablok regressions demonstrated some differences between the POCT and reference method, which are depicted in Table 1 and visualised in Figure 1. The comparison of IFX CWB POCT and IFX ELISA demonstrated a systematic bias, as the 95% CI of the intercept did not enclose 0 (intercept 1.56, 95% CI 1.10–1.93), but no longitudinal bias was detected, since the 95% CI of the slope enclosed 1 (slope 1.04, 95% CI 0.96–1.11), showing overestimation of IFX concentration when measured with CWB POCT. There was also systemic bias when five suspected timing errors were excluded (intercept 1.67, slope 1.01 (1.32–2.05). The IFX serum POCT showed less systemic bias (intercept 0.71, 95% CI 0.37–1.07), but did show conditional bias (slope 1.10, 95% CI 1.05–1.16) when compared with the IFX ELISA. ADL CWB POCT had a systemic bias in the intercept (1.44, 95% CI 0.73–2.14), but no longitudinal bias in the slope (1.09, 95% CI 0.98–1.21). There was conditional and systemic bias for the CRP and FCP comparison.

Considering the ICC, there was almost perfect agreement between the reference method and the IFX CWB POCT, IFX serum POCT, ADL CWB POCT and CRP CWB POCT (Table 2). There was moderate agreement with the FCP Elia and POCT. Pearson’s and Spearman’s Rank correlation demonstrated strong associations between POCT results and the reference methods.

Bland–Altman plots showed a negative bias for IFX serum and CWB POCT and ADL CWB POCT, meaning the POCT systematically overestimated the anti-TNF ELISA concentrations with 1.6–2.3 µg/mL (Figure 2). Using the POCT device, there was an underestimation of the CRP and FCP reference method results. In all comparisons, the dispersion of the difference between the reference and POCT became more prominent when the results were higher. For anti-TNF concentrations, this dispersion existed especially in values higher than 8 µg/mL.

The proportion of false positive and false negative results are demonstrated in Table 3 with the corresponding NPV and PPV. For some participants, there were significant differences in the comparison of IFX ELISA and IFX CWB POCT results (Appendix A). Four of the largest outliers between the IFX ELISA and IFX CWB POCT were 13.2, −14, −31 and −70.2 µg/mL. The differences from these corresponding patient results were 6.2, −1.7, −0.9 and −2.2, respectively, when the IFX ELISA was compared with IFX serum POCT results.

## 4. Discussion

Here, we present a validation study of a novel POCT device that was used for IBD patients to measure IFX and ADL serum concentrations using CWB, as well as serum CRP and FCP. The POCT device was easy to use and rapid, since the result were received within 5 min and did not require technically difficult steps. The POCT slightly overestimated IFX and ADL ELISA results. Good predictive properties were observed for predicting a value above or below clinically relevant values. The agreement was comparable to previous studies using a less user-friendly IFX POCT device [9,10,18,19,20]. Additionally, the ADL ELISA and ADL CWB POCT comparison showed similar agreement as previous literature [14,20,21,22]. CRP and FCP POCT slightly underestimated CRP and FCP concentrations, and this difference became more prominent when CRP and FCP concentrations were in the higher range. In line with this notion, previous studies comparing different IFX, ADL, CRP and FCP assays found heterogeneity to some extent [9,20,22,23,24]. Therefore, it is important to follow up certain test results with caution in case another assay has been used for previous measurements. POCT results somewhat above or below a certain clinically relevant threshold should also be carefully interpreted.

The IFX POCT was also repeated using corresponding serum samples as outliers were observed in which the IFX CWB POCT was considerably higher than the IFX ELISA. We hypothesised that these large differences were caused by timing errors in which the venous blood withdrawal was executed before the IFX infusion was started and, by mistake, the finger prick for a CWB sample was performed while the IFX infusion was running in. Indeed, this was confirmed as the highest outliers between the IFX ELISA and IFX CWB POCT were absent when IFX POCT was repeated using residual serum samples, when available. In the additional analysis of IFX CWB POCT in which five suspected timing errors were excluded, agreement with IFX ELISA was stronger. Interestingly, the IFX serum POCT showed higher overall agreement with IFX ELISA than the CWB POCT. Apart from these timing errors, we were not able to identify other associations for high variability between the reference test and POCT. No trend was observed for these outliers and the study professional who performed the test, patient characteristics or a certain day or time on which tests were performed. To improve the test accuracy, it was important to follow the manufactures instructions. This included that the serum or faecal sample had to be mixed with the buffer bulb by vigorously inverting this five times, and to minimize the time between taking the sample and performing the test.

The FCP POCT showed only moderate agreement with the FCP Elia results. Nevertheless, these results of the FCP POCT in our study are in line with a comparison study of six different FCP assays which also showed considerable inter-test variability [24]. Calprotectin is a protein which makes up 60% of the total cytosolic protein in neutrophils [25]. Hence, calprotectin is more abundant in stool samples where blood cells are present, causing high test variability.

The present study had some strengths. We recruited a relatively high number of patients in a standard clinical setting, especially for the IFX and ADL validation. Gaining experience on the feasibility of a new device in a clinical setting is vital to implement it in standard care. This study also has some limitations. We used only one reference test for each POCT and did not perform tests on standard samples, since this has already been conducted [11,12,13]. For the FCP validation, only 33 participants had a FCP POCT result within the assay range, whereas we intended to include 40 participants with both test results within the assay ranges. We did not have corresponding endoscopic data, since patients were recruited in routine care. As this was a validation study, we did not use the POCT results for clinical decision-making. Future studies should focus on the clinical implications of POCT results on IBD management and whether discrepancies exist with decision-making based on the conventional reference methods.

In conclusion, this novel POCT device was fast, user-friendly and reliable for clinical purposes. Overall, IFX and ADL results were slightly higher with the POCT, whereas CRP and FCP results were slightly lower than the reference methods. Large differences between the two methods mainly occurred in higher ranges, making this less clinically relevant. The POCT device has the potential to be implemented in a clinical or research setting, enabling rapid decision-making in IBD management.

## Figures and Tables

**Figure 1 diagnostics-13-01712-f001:**
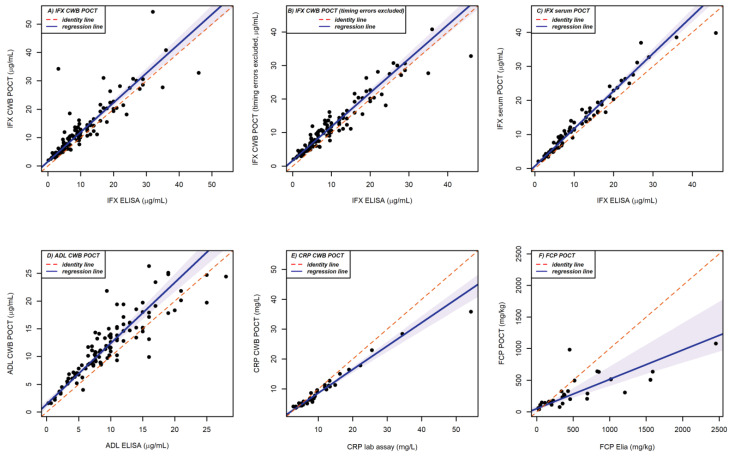
**Passing–Bablok regressions**. Comparison of the reference method with A: IFX CWB POCT, B: IFX CWB POCT with five suspected timing errors excluded, C: IFX serum POCT, D: ADL CWB POCT, E: CRP CWB POCT and F: FCP POCT. Black dots represent individual values, the dashed red line represents the ‘perfect’ agreement and the grey line is the regression line surrounded by the 95% confidence interval in shaded grey.

**Figure 2 diagnostics-13-01712-f002:**
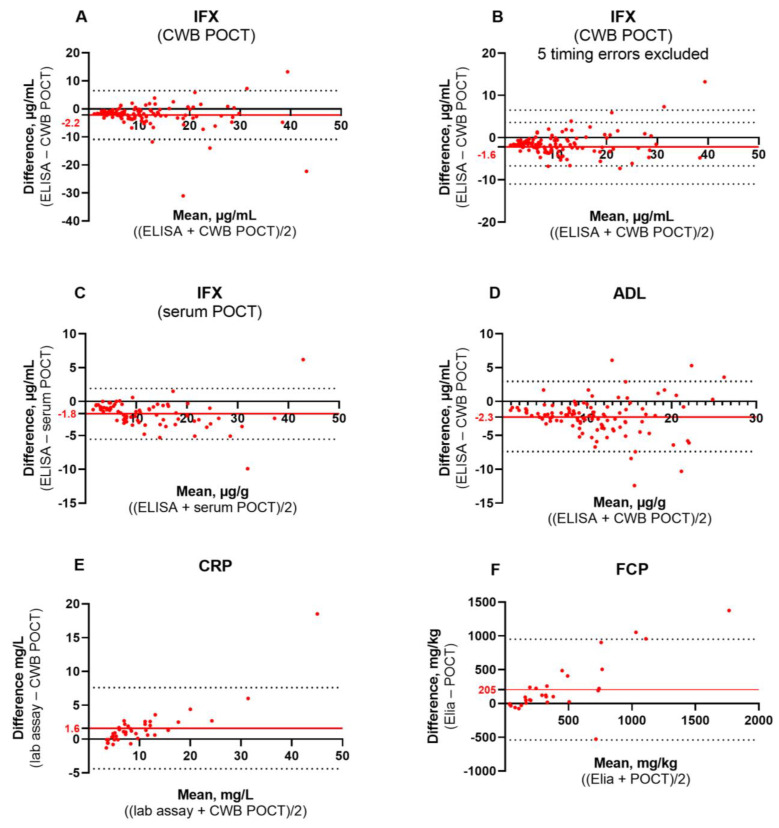
**Bland-Altman plots**. Comparison of the reference method with (**A**): IFX CWB POCT, (**B**): IFX CWB POCT with five suspected timing errors excluded, (**C**): IFX serum POCT, (**D**): ADL CWB POCT, (**E**): CRP CWB POCT and (**F**): FCP POCT. The *y*-axis shows the differences between each test result and the *x*-axis shows the mean of each test result. The red horizontal line represents the bias (the mean difference between the reference method and POCT), with the absolute number of the bias depicted on the *y*-axis in red font. Dashed black horizontal lines represent the upper and lower limits of the bias. Red dots represent each individual participant.

**Table 1 diagnostics-13-01712-t001:** Baseline characteristics.

Participants total * (*n*)	285
IFX CWB POCT (*n*)	124
IFX trough (*n*)	103
IFX intermediate (*n*)	21
IFX serum POCT (*n*)	79
ADL CWB POCT (*n*)	115
Participants sampled 2 times (*n*)	22
Participants sampled 3 times (*n*)	1
CRP (*n*)	65
FCP (*n*)	50
Female (*n*, %)	173 (60.7%)
Age, mean (SD)	41.6 (16.8)
Crohn’s disease (*n*, %)	207 (72.6%)
Disease duration (years, SD)	14.7 (13.0)
Ulcerative colitis (*n*, %)	78 (27.4%)
Disease duration (years, SD)	11.7 (10.3)

* Some patients participated more than once in the ADL comparison or to multiple comparisons.

**Table 2 diagnostics-13-01712-t002:** Passing–Bablok regression and intra-class, Pearson and Spearman’s rank correlations.

Reference Method	POCT	*n*	*n*, Both Results within Assay Range	Passing–Bablok Regression		ICC (95% CI)	Pearson Correlation	Spearman’s Rank Correlation
				Intercept (95% CI)	Slope (95% CI)			
IFX ELISA	IFX CWB POCT	124	120	1.56 (1.10–1.93)	1.04 (0.96–1.11)	0.85 (0.73–0.91) *	0.88 *	0.91 *
IFX ELISA	IFX CWB POCT **	119	116	1.67 (1.32–2.05)	1.01 (0.94–1.07)	0.93 (0.84–0.96) *	0.95 *	0.96 *
IFX ELISA	IFX serum POCT	79	76	0.71 (0.37–1.07)	1.10 (1.05–1.16)	0.96 (0.74–0.96) *	0.98 *	0.99 *
ADL ELISA	ADL CWB POCT	115	110	1.44 (0.73–2.14)	1.09 (0.98–1.21)	0.82 (0.43–0.92) *	0.89 *	0.92 *
CRP lab assay	CRP CWB POCT	65	41	0.81 (0.22–1.40)	0.78 (0.71–0.85)	0.91 (0.80–0.96) *	0.98 *	0.96 *
FCP Elia	FCP POCT	50	33	51 (21–92)	0.46 (0.35–0.69)	0.55 (0.22–0.76) *	0.78 *	0.95 *

* *p* < 0.001, ** five patients with timing errors were excluded in this comparison.

**Table 3 diagnostics-13-01712-t003:** Discrepancies above and below clinically relevant thresholds.

Reference Method	POCT	Discrepancies *n*/N (%)	Interpretation POCT	NPV	PPV
IFX ELISA > 3 µg/mL	IFX CWB < 3 µg/mL	0/124 (0%)	IFX CWB false negative	100%	88.2%
IFX ELISA < 3 µg/mL	IFX CWB > 3 µg/mL	14/124 (11.3%)	IFX CWB false positive		
Five timing errors excluded					
IFX ELISA > 3 µg/mL	IFX CWB < 3 µg/mL	0/120 (0%)	IFX CWB false negative	100%	87.7%
IFX ELISA < 3 µg/mL	IFX CWB > 3 µg/mL	14/119 (12.3%)	IFX CWB false positive		
IFX ELISA > 3 µg/mL	IFX serum < 3 µg/mL	0/79 (0%)	IFX serum false negative	100%	89.5%
IFX ELISA < 3 µg/mL	IFX serum > 3 µg/mL	5/79 (6.3%)	IFX serum false positive		
ADL ELISA > 3 µg/mL	ADL CWB < 5 µg/mL	1/115 (0.9%)	ADL CWB false negative	93.3%	89.0%
ADL ELISA < 3 µg/mL	ADL CWB > 5 µg/mL	11/115 (9.6%)	ADL CWB false positive		
CRP lab assay > 5 mg/dL	CRP CWB < 5 mg/dL	2/65 (3.1%)	CRP CWB false negative	90.9%	96.9%
CRP lab assay < 5 mg/dL	CRP CWB > 5 mg/dL	1/65 (1.5%)	CRP CWB false positive		
FCP Elia > 250 µg/mg	FCP Elia < 250 µg/mg	5/50 (10%)	FCP CWB false negative	85.3%	100%
FCP Elia < 250 µg/mg	FCP Elia > 250 µg/mg	0/50 (0%)	FCP CWB false positive		

NPV = negative predictive value, PPV = positive predictive value.

## Data Availability

Request for data can be made to the corresponding author. The study team will decide on the decision to share the requested data.

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
