# Peer review of "Validation Study of Novel Point-of-Care Tests for Infliximab, Adalimumab and C-Reactive Protein in Capillary Blood and Calprotectin in Faeces in an Ambulatory Inflammatory Bowel Disease Care Setting"

_diagnostics, 2023, doi:10.3390/diagnostics13101712_

Round 1
Reviewer 1 Report
The authors presented a validation study of a novel POCT device that was used for IBD patients to measure IFX, ADL, CRP and FCP. The results showed that IFX and ADL results were slightly higher with the POCT, whereas CRP and FCP results were lower than the reference methods. Here are the comments:
1. The authors did not analyze the reasons for the variability of the results of POCT assays from those of conventional assays on the stand of its underlying mechanism, especially for CRP and FCP, where the variability is relatively large.
2. The authors may consider adding a POCT test performance on standard samples.
3. The table in the article is not a trilinear table
4. It would be better to add subheadings to the images in the article, as well as the statistical data corresponding to the independent images, for easy viewing by readers.
Author Response
The authors presented a validation study of a novel POCT device that was used for IBD patients to measure IFX, ADL, CRP and FCP. The results showed that IFX and ADL results were slightly higher with the POCT, whereas CRP and FCP results were lower than the reference methods. Here are the comments:
Authors response: We would like to thank the reviewer for these comments.
- The authors did not analyze the reasons for the variability of the results of POCT assays from those of conventional assays on the stand of its underlying mechanism, especially for CRP and FCP, where the variability is relatively large.
Authors response: We identified that, in some cases, the finger prick for IFX CWB POCT was withdrawn while the infliximab infusion was already running in. The serum sample had been withdrawn prior to the infusion. Hence, the POCT results were much higher than the ELISA results. We present the IFX comparison data both with all data points and with five patients excluded in which we clearly observerd that there was such a timing error. Apart from these timing errors, we were not able to identify any reasons for the variability like the person who performed the test or a certain day or time. We have added a sentence in the discussion (page 13) section to make this point more clearly. However, we describe in the discussion section that for FCP tests in general, the variability is large.
- The authors may consider adding a POCT test performance on standard samples.
Authors response: In the discussion section (page 14, line 290) we refer to previous publications in which the test performances on standard samples are described. The aim of our study was to test this novel device’s accuracy in a standard clinical setting
- The table in the article is not a trilinear table
Authors response: Something went wrong with the outlining of the tables. We corrected the proof and will revise the proof carefully in case the manuscript is accepted.
.
- It would be better to add subheadings to the images in the article, as well as the statistical data corresponding to the independent images, for easy viewing by readers.
Authors response: We changed the figure legends accordingly. With regard to the figures and tables, we made sure they are easy to read. We will check this in the proof version.
Reviewer 2 Report
The authors have done a very good work and can be accepted in the journal. The authors are need to modify the resolution of the images and few grammatical mistakes.
Author Response
The authors have done a very good work and can be accepted in the journal. The authors are need to modify the resolution of the images and few grammatical mistakes.
Authors response: We would like to thank the reviewer for these comments and the positive feedback. We have carefully reviewed the manuscript for grammatical mistakes. We will upload the images in high quality and in case of acceptance, we will modify the layout of the images and tables.
Reviewer 3 Report
This study aims to validate this novel Point-of-Care tests device by comparing results with reference methods for infliximab (IFX), adalimumab (ADL) serum concentrations, C-reactive protein (CRP) and faecal calprotectin (FCP) in a clinical inflammatory bowel disease setting. The experiments are well-designed and provide evaluation of the Point-of-Care tests device.
The introduction of the IFX, ADL,CRP and FCP monitoring in IBD treatment can be improved.
In the discussion part, the author can include the possible methods to improve the accuracy of the Point-of-Care tests device.
Author Response
This study aims to validate this novel Point-of-Care tests device by comparing results with reference methods for infliximab (IFX), adalimumab (ADL) serum concentrations, C-reactive protein (CRP) and faecal calprotectin (FCP) in a clinical inflammatory bowel disease setting. The experiments are well-designed and provide evaluation of the Point-of-Care tests device.
Authors response: We would like to thank the reviewer for these comments and the positive feedback.
The introduction of the IFX, ADL,CRP and FCP monitoring in IBD treatment can be improved.
Authors response: We carefully reviewed the introduction section of our manuscript and improved the last paragraph of the introduction.
In the discussion part, the author can include the possible methods to improve the accuracy of the Point-of-Care tests device.
Authors response: Based on our data, we are not able to make strong claims on how to improve accuracy. It is important to follow the manufactures instructions as described in the methods section. We added a sentence on page 13/14 line 280-282.
Round 2
Reviewer 3 Report
Thanks for the authors' reply. My questions were well addressed.
Author Response
thank you